# A Current Overview of Cyclodextrin-Based Nanocarriers for Enhanced Antifungal Delivery

**DOI:** 10.3390/ph15121447

**Published:** 2022-11-22

**Authors:** Hay Man Saung Hnin Soe, Phyo Darli Maw, Thorsteinn Loftsson, Phatsawee Jansook

**Affiliations:** 1Faculty of Pharmaceutical Sciences, Chulalongkorn University, 254 Phyathai Road, Pathumwan, Bangkok 10330, Thailand; 2Faculty of Pharmaceutical Sciences, University of Iceland, Hofsvallagata 53, IS-107 Reykjavik, Iceland

**Keywords:** cyclodextrins, nanotechnology, solubilization, antifungal, drug delivery, bioavailability

## Abstract

Fungal infections are an extremely serious health problem, particularly in patients with compromised immune systems. Most antifungal agents have low aqueous solubility, which may hamper their bioavailability. Their complexation with cyclodextrins (CDs) could increase the solubility of antifungals, facilitating their antifungal efficacy. Nanoparticulate systems are promising carriers for antifungal delivery due to their ability to overcome the drawbacks of conventional dosage forms. CD-based nanocarriers could form beneficial combinations of CDs and nanoparticulate platforms. These systems have synergistic or additive effects regarding improved drug loading, enhanced chemical stability, and enhanced drug permeation through membranes, thereby increasing the bioavailability of drugs. Here, an application of CD in antifungal drug formulations is reviewed. CD-based nanocarriers, such as nanoparticles, liposomes, nanoemulsions, nanofibers, and in situ gels, enhancing antifungal activity in a controlled-release manner and possessing good toxicological profiles, are described. Additionally, the examples of current, updated CD-based nanocarriers loaded with antifungal drugs for delivery by various routes of administration are discussed and summarized.

## 1. Introduction

Fungal infections are prevalent causes of morbidity and mortality in humans. The prolonged use of antibiotics or immunosuppressive agents leads to the increased vulnerability of human beings and renders them prone to fungal infections. Infections caused by opportunistic fungi or other pathogens are classified as superficial, cutaneous, subcutaneous, mucosal, or systemic based on the severity [1,2]. The most common pathogens causing life-threatening fungal infections are *Candida* spp., *Aspergillus* spp., *Fusarium* spp., *Cryptococcus* spp., and *Pneumocystis* spp. They mostly relate to immunocompromised patients, such as immunodeficient patients receiving immunosuppressive therapy or patients with cancer undergoing chemotherapy [3,4]. 

Globally, almost one-fourth of the human population is affected by *Candida*. Candidiasis is the most common systemic and localized infection among hospitalized patients worldwide [5]. Among the *Candida* infections, oral candidiasis and vaginal candidiasis are the most common infections [6,7]. The occurrence rate of oral candidiasis has significantly risen with the use of immunosuppressive drugs and immunodeficiency diseases [8]. The reported morbidity and mortality rates of invasive fungal infections are 1.5 million patients yearly. According to a survey, 90% of deaths resulting from fungal infections are caused by *Candida*, *Cryptococcus*, *Aspergillus*, and *Pneumocystis* [9]. In certain geographic regions, *Blastomyces*, *Coccidioides*, *Histoplasma*, *Talaromyces*, *Paracoccidioides*, and *Sporotrichosis* are the fungal pathogens causing life-threatening endemic mycoses [1,10]. 

At present, clinically-applied antifungal drugs comprise five categories, i.e., azoles, polyenes, echinocandins, allylamines, and pyrimidine analogs, which are used to treat superficial and systemic fungal infections [11,12]. The antifungal activity of azoles (imidazole or triazoles antifungals) is achieved by inhibiting the sterol 14α-demethylase, an essential enzyme required for sterol biosynthesis, converting lanosterol to ergosterol, thereby destroying the stability and fluidity of fungal cell membranes. Amphotericin B, nystatin, and natamycin, which are polyene antifungal agents, bind to the fungal cell membrane and lead to pore formation and cell death. The echinocandins (caspofungin, micafungin, and anidulafungin) are semisynthetic compounds that block fungal cell wall synthesis, whereas allylamine antifungals, i.e., terbinafine and naftifine, inhibit fungal growth by obstructing squalene epoxidase, an enzyme required for ergosterol biosynthesis [2,13]. Another class of antifungal drugs, i.e., pyrimidine analogs (5-fluorocytosine and 5-fluorouracil), exhibit antifungal activity by blocking DNA and RNA protein synthesis in fungal cell membranes [14]. The brief mechanisms of the actions of antifungals are illustrated in Figure 1.

Almost all of the above antifungals possess limitations due to their physicochemical properties, especially low aqueous solubility and permeability, leading to poor bioavailability and clinical efficacy [15,16]. Another critical factor is that, to achieve effective therapeutic action, a sufficient drug concentration at the site of infection is necessary. To address these issues, researchers have developed formulations or nanocarrier systems incorporating cyclodextrins (CDs). CDs are well-known cyclic oligosaccharides that can form water-soluble inclusion complexes with lipophilic compounds [17]. The solubility enhancement of poorly water-soluble drugs and the permeability enhancement of drugs through biological membranes are attained via the formation of drug/CD inclusion complexes [18]. Thus, CDs could facilitate the drug efficacy and bioavailability of many antifungal drugs [19,20,21]. In this present review, we focused on the updated information of currently available antifungal drug-loaded CD-based nanocarriers or formulations that are applied in different drug delivery systems to treat various fungal diseases effectively.

## 2. CDs as Solubilizers, Penetration Enhancers, and Stabilizers 

CDs are obtained from the enzyme-catalyzed starch degradation by glucosyltransferase enzymes [22]. Their molecular structures resemble a torus-like molecular ring in which the inner cavity is hydrophobic and the outer one is hydrophilic. The hydroxyl functional groups are orientated to the outer cavity, whereas the interior cavity is lined with skeletal carbons and the ethereal oxygens of the glucose residues, representing its lipophilic character (Figure 2). The primary hydroxyl groups are located at the narrow rim of the torus, whereas the secondary hydroxyl groups are positioned at the wide rim of the CD structure. CDs have the ability to enhance the aqueous solubility of various poorly water-soluble drugs by inserting the lipophilic moiety of the drugs into their central hydrophobic cavity [17]. The parent CDs, α-cyclodextrin (αCD), β-cyclodextrin (βCD), and γ-cyclodextrin (γCD), possess six, seven, and eight glucopyranose units, respectively. In the CD molecules, it is possible to form intramolecular hydrogen bonds between the C2-OH of one glucopyranose unit and the C3-OH of an adjacent glucopyranose unit. Particularly in βCD, a complete secondary belt formation exists that leads to its molecular rigidity, thereby reducing the ability of βCD to form hydrogen bonds with surrounding water molecules. For this reason, βCD has the lowest water solubility among the parent CDs. In contrast, one glucopyranose unit of αCD is in a distorted orientation with an incomplete hydrogen bond belt and enhanced solubility compared to βCD. γCD possesses a noncoplanar with a more flexible structure and, thus, is more soluble in water than both αCD and βCD [23,24]. Random substitution at the hydroxy groups located on the outer surface of the CD molecule dramatically improves the solubility of CDs (Table 1).

### 2.1. Cyclodextrin as a Solubilizer

Poorly water-soluble drugs have an affinity to form complexes with CD in an aqueous solution [26]. As mentioned above, the hydrophobic character of the inner CD cavity enables it to accommodate the lipophilic compound in its inner cavity via the inclusion complex formation, and hence the aqueous solubility of the drug is significantly enhanced. Generally, no formation or breakage of covalent bonds occurs between the drug and CD molecules during the complex formation. In aqueous media, the complexes are in dynamic equilibrium with free molecules, that is, the free drug and CD molecules. The driving forces of the complexation include the release of water molecules from the cavity, electrostatic interactions, van der Waals’ interactions, hydrophobic interactions, hydrogen bonding, the release of conformational and steric strains, as well as charge-transfer interactions [17,31]. An improved drug solubility results in enhanced drug bioavailability and, consequently, improved therapeutic efficacy. The CD solubilizing effects on antifungal agents, reported from the literature, are summarized in Table 2.

The CD solubilization of hydrophobic drugs is most frequently evaluated using the phase-solubility technique, according to Higuchi and Connors [56]. The two main types of diagrams are type A (i.e., soluble complexes) and type B (i.e., complexes with limited solubility). Type A diagrams are divided into three subtypes: linear phase-solubility diagrams (A_L_), phase-solubility diagrams with a positive deviation from linearity (A_P_), and phase-solubility diagrams with a negative deviation from linearity (A_N_). The A_L_ type is generally observed when one drug molecule fits into one CD cavity (1:1), while the A_P_ type indicates that one drug molecule forms a complex with two CD molecules (1:2). The type B diagrams are divided into two subtypes: phase-solubility diagrams where the complex has some but limited solubility (B_S_), and diagrams where the complex is insoluble in the complexation medium (B_i_). The complexation depends on various factors, such as the molecular structure of the guest molecule, the size of the CD cavity, the complexation medium, the preparation method, the experimental temperature, etc. 

For example, amphotericin B, a large polyene antifungal compound, has the highest affinity for the γCD cavity among the parent CDs [48]. As the experimental temperature was increased, better complexation was observed, for example, in the case of posaconazole/βCD (Table 2) [46]. In addition, the pH of the aqueous medium is also important when the guest molecule is ionizable. Econazole is a small molecule that fits into the αCD cavity. Due to its pH-dependent ionization (pKa = 6.6), the CE of econazole/αCD was higher in the complexing medium with a pH of 3, followed by pH 5 and 7.5 [36]. The solubility of itraconazole also strongly depends on the pH of aqueous solutions. Due to the ionizable nature of itraconazole (pKa = 3.7) and the fact that it possesses four ionizable nitrogen atoms in its structure, the pH-dependent solubility was prominent in aqueous HPβCD solutions. The 1:1 stoichiometry was dominant in a low pH medium (pH 2) due to a fully protonated state, whereas the higher potential of a 1:2 complex formation was observed in a pH 4 medium (partial ionization) state [42]. Generally, the unionized form of a drug molecule has a higher K_1:1_ value than the ionized form due to a greater affinity for the lipophilic CD cavity. However, in the case of greatly increased intrinsic solubility from the drug ionization, an overall increased CE could be obtained. In most cases, the heating of aqueous drug suspensions for the preparation of antifungal/CD complexes could enhance the CE. To the best of our knowledge, heating promotes supersaturated drug solutions and increases the solubility of natural CDs, leading to improved complex formation [57].

### 2.2. Cyclodextrin as a Permeation Enhancer

Penetration enhancers, i.e., fatty acids, ethanol, surfactants, bile salts, and polymers, such as chitosan, enhance the permeation of drugs through intranasal, ocular, buccal, transdermal, and intestinal mucosa. They increase the permeation of drugs across the mucous membranes by directly penetrating lipophilic membranes, where they decrease the membrane barrier properties by enhancing the hydration and structural changes of the membrane [58,59,60,61]. 

In lipophilic biological membranes, there is a layer of a cell-based epithelium, and above it is a stagnant aqueous diffusion barrier that is lined with mucous. This barrier is a rate-limiting step for the penetration of lipophilic drugs through membranes [62]. The permeability of a molecule through a lipophilic membrane depends on the molecular weight, structure, and physicochemical properties of the drug. Fick’s first law states that the rate of passive drug permeation through a membrane barrier surface is proportional to the concentration gradient of a dissolved drug. Hence, the drug must have not only sufficient lipophilicity allowing for drug partition into and then permeation through the lipophilic membrane, but also adequate solubility in the aqueous exterior layer to diffuse to the lipophilic membrane surface. 

The use of CDs as penetration enhancers has become popular due to their distinct characteristics. They can form complexes with drugs, dissociate and disaggregate proteins, are compatible with many biological tissues, and are commercially available. The mechanism of drug permeation enhancement through biological membranes is mainly passive diffusion. By forming complexes, CDs can carry and deliver drugs through the aqueous barrier toward the lipophilic surfaces of biological membranes and allow the permeation of the drug molecules from the complex into the lipophilic membrane [18]. However, CDs do not readily pass through lipophilic biological membranes, according to Lipinski’s rule-of-five. That is, CD molecules contain a large number of hydrogen bond donors and acceptors, have a relatively high molecular weight (from 900 to over 2000 Da), and have high negative octanol/water partition coefficients (Log P_o/w_) [63,64,65,66,67]. Only free drug molecules are able to permeate lipophilic membranes. Drug/CD complexes increase drug availability immediately to the lipophilic membrane surface, where free drug molecules partition into and then permeate through the membrane [67,68]. The reported studies regarding CDs enhancing the permeability of antifungal drugs through mucous membranes are summarized in Table 3.

### 2.3. Cyclodextrin as Stabilizer

Most active pharmaceutical ingredients are prone to chemical degradation in aqueous media and do so by various pathways, such as hydrolysis, oxidation, photodegradation, and isomerization. Because of degradation, the therapeutic efficacy of drugs is decreased. The chemical stability of drugs and the shelf lives of pharmaceutical products could be increased by forming CD inclusion complexes [71]. 

Nystatin is a polyene antifungal agent widely used for the topical treatment of fungal infections. However, the major limitations of its use are its poor solubility and chemical instability in aqueous solutions, making it prone to photodegradation, hydrolysis, thermal degradation, and oxidation [72]. Doorne and Bosch (1991) [73] showed that γCD enhanced the stability of nystatin by encapsulating the polyene portion (the degradable part of nystatin) inside the γCD cavity, thereby successfully retarding the oxidation and photodegradation of nystatin. A similar finding was reported by Wang et al. (2014) [51]. In their findings, the Fourier transform infrared spectrum of a nystatin/γCD complex exhibited that the labile parts of nystatin, i.e., the diene and ester groups, were included in the γCD cavity. This information supported γCD’s ability to protect nystatin against degradation. The nystatin/γCD inclusion complexes stored in an incubator under light significantly improved the stability of nystatin at 4 °C and 25 °C, whereas at 50 °C, some degradation was observed. 

The stability of another polyene antifungal agent, natamycin, was also improved by using CD complex formation. Natamycin is known to be prone to degradation by extreme pH conditions, heat exposure, and oxidation [74,75]. Natamycin/βCD and natamycin/γCD complexes were more stable than intact natamycin in aqueous solutions stored in the dark at 4 °C [76]. Posaconazole, an analog of itraconazole, is more susceptible to oxidative degradation than other degradation mechanisms [77]. In addition, it could generate three degradation products under oxidative stress conditions due to the insertion of oxygen atoms in the piperazine ring located in the middle part of the structure [78]. The degradation kinetics of posaconazole were studied in the absence and presence of βCD under oxidative stress conditions induced by hydrogen peroxide (H_2_O_2_). The encapsulation of βCD at both ends of posaconazole prevented the degradation of posaconazole by hindering the liable sites that are attacked by H_2_O_2_ molecules.

## 3. CD-Based Nanocarriers in Enhanced Antifungal Delivery

Presently, nanotechnology offers an ingenious solution for overcoming the drawbacks of poorly water-soluble drugs in formulation development. The use of several CD-based nanocarriers in formulations has been reported for drug delivery systems (Figure 3). The combined technology has also been of interest as an effective system for antifungal drug delivery. The use of biocompatible and biodegradable components, as well as versatility, stability, high drug loading, and permeability through biological membranes, are obvious benefits of using these carrier systems for the effective delivery of antifungal agents. 

The following are case studies of CD-based formulations for antifungal drug delivery. The effects of CDs on enhancing the solubility and permeability of drugs through various routes of administration, leading to enhanced bioavailability and therapeutic antifungal activity, are also discussed.

### 3.1. Oral Drug Delivery

Oral dosage forms are recognized as the most acceptable formulations for patients. However, due to low aqueous solubility, poor permeability, chemical instability, degradation in the gastrointestinal tract, extensive metabolism, and the unpleasant organoleptic properties of drugs, the oral route is still challenging for developing pharmaceuticals. To overcome these drawbacks, CDs were used to increase the solubility and enhance the dissolution rate of poorly water-soluble drugs [79].

Kumar et al. (2014) [80] studied the effect of CD on the solubility enhancement of itraconazole. Firstly, itraconazole sulfate salt was synthesized and prepared in physical mixtures with βCD and HPβCD. The physical mixtures resulted in an enhanced dissolution rate in simulated gastric fluid (pH 1.2) compared with itraconazole salt and pure itraconazole. Of these, the dissolution rate of the mixture with HPβCD was higher than that of βCD. The authors suggested that the developed itraconazole sulfate/CD mixtures could be potentially applied to oral drug delivery systems.

Floating drug delivery systems are one of the most widely used approaches to increase the gastric retention time of oral dosage forms. Among the single and multiple-unit systems, multiple-unit particulate dosage forms containing microspheres show more benefits due to their uniformity when passing along the gastrointestinal tract, preventing gastric emptying and providing controlled release, thereby decreasing the variability in drug absorption and reducing local irritation [81]. Itraconazole, an orally administered antifungal compound, was loaded in microspheres using an ionotropic gelation technique with chitosan (1.5–2.5% *w*/*v*). The resulting microspheres containing 1% *w*/*v* itraconazole were lyophilized and further developed into floating tablets. The in vitro permeation was studied at different pH values, i.e., pH 5, 6, and 7.4. The results revealed that the highest permeability was observed from a formulation composed of 4.72% *w*/*v* itraconazole/RMβCD, 2% *w*/*v* chitosan, 3% *w*/*v* polyethylene glycol, and 1% *w*/*v* tripolyphosphate at a pH of 5. The optimized formulation was evaluated for in vivo buoyancy activity with gamma scintigraphy using rabbits and showed it remained floating in the stomach for 6.5 h and had improved bioavailability [20]. 

Due to its poor aqueous solubility, posaconazole has been administered as an oral suspension. To enhance the solubility of posaconazole, a posaconazole/HPβCD inclusion complex was prepared and characterized by Tang et al. (2016) [47]. The solubility of posaconazole was greatly enhanced (i.e., 82 times), and a high dissolution (>90%) was obtained when complexed with HPβCD at different pH levels representing gastrointestinal tract conditions. The in vitro antifungal susceptibility study also proved that complex was the most susceptible to *Candida albicans*, and nearly 97% of fungi were inhibited by ≤1 µg/mL under the minimal inhibitory concentration (MIC) at which 90% of the isolates were inhibited (MIC90) with 0.06 µg/mL. In the case of the least susceptible *Aspergillus niger*, 97% of the MICs were ≤4 µg/mL, and the MIC90 was at 1 µg/mL. Their findings revealed that HPβCD was able to enhance the aqueous solubility and dissolution rate in simulated gastric conditions, thereby maintaining a high susceptibility to the tested fungal species. 

### 3.2. Oral Local Drug Delivery

In the oral cavity, candidiasis is the most common fungal infection. Its clinical manifestations range from relatively minor disorders, such as acute pseudomembranous forms (e.g., oral thrush), to chronic erythematous and hyperplastic forms, as well as systemic superinfections among immunocompromised patients, with mortality rates of 30 to 50% [82]. Despite the availability of various effective antimycotics to treat oral candidiasis, therapeutic failure is common due to the oral cavity’s peculiar features, tending to lower local drug concentrations to subtherapeutic levels [83]. 

Solid sponge-like matrices (wafers) obtained by lyophilizing polymer solutions offer advantages over conventional drug delivery systems; for example, they can maintain their swelling gel structure for extended periods of time within the oral cavity [84]. In the study of Mura et al. (2015) [85], they prepared an econazole/SBEβCD/citric acid ternary complex to improve the solubility of econazole. The binary, as well as the ternary complexes, provided a significantly higher antimycotic efficacy against selected *Candida* strains compared with that of econazole alone. Again, the ternary complex, providing high solubility as well as dissolution efficiency, was selected and loaded into low methoxy amidated pectin and carboxymethylcellulose to develop a lyophilized wafer formulation. The therapeutic antifungal efficacy using a time-killed assay was also tested in vitro for *Candida albicans* and *Candida krusei*, and resulted in 30.99 ± 1.23 h and 33.01 ± 4.01 h, respectively. The developed wafer formulation did not affect or lessen the antifungal activity of econazole when compared with that of the ternary system. 

Antifungals are also delivered by their incorporatiation into medicated chewing gum, which is a convenient delivery system for fungal infections in the oral cavity [86]. Econazole and miconazole solubilities were enhanced by complexing with CDs [87]. Then, both the drug/CD complex, as well as free drugs, were incorporated into the chewing gum, which was prepared by mixing a Fertin gum base, solid paraffin wax, and other excipients. In the case of econazole, the econazole/βCD complex only moderately increased the release of econazole compared with the release from the neat econazole gum. The miconazole/HPβCD complex-loaded gum had a much higher drug release (25% within 30 min) than the miconazole gum. When compared with neat miconazole chewing gum, miconazole/HPβCD chewing gum demonstrated superior antifungal activity due to the increased drug release. 

For decades, advanced research on drug delivery systems has focused on enhancing therapeutic effects. Nanofibers have also gained attention in drug delivery because of their numerous advantages, including their high loading and encapsulation of drugs and great flexibility for incorporating a wide range of materials [88,89]. Fast-dissolving nanofiber mats were optimized using a polyvinyl pyrrolidone (PVP)/HPβCD nanofiber for clotrimazole delivery in the oral mucosa [90]. CD could increase drug dissolution in the saliva, thereby facilitating drug absorption and resulting in enhanced drug bioavailability. For treating oral candidiasis, both rapid antifungal activity and prolonged contact time to reach the MIC are required. Because the resultant PVP/HPβCD nanofibers displayed fast drug release, modified coated nanofibers with chitosan and polyvinyl alcohol were employed to increase the contact time with the oral mucosa. The modified clotrimazole nanofibers demonstrated good mucoadhesive and controlled-release properties and antifungal activity against *Candida albicans* [91]. 

### 3.3. Vaginal Drug Delivery 

In recent years, vaginal infections have become the most common gynecologic disorder in which various pathogens, such as fungi, viruses, and protozoa, cause various diseases in the female reproductive tract. Among the pathogenic fungi, *Candida albicans* is still considered the most prevalent fungal pathogen responsible for vulvovaginal candidiasis. Because of the gastric pH and the extensive first-pass metabolism for drugs taken orally, as well as to avoid systematic side effects, the vaginal route has been preferred to administer antifungals for local therapy [92]. However, the dynamic elimination mechanisms in the vaginal lumen cause insufficient residence time at the target absorption site, causing an uneven distribution of drugs onto the vaginal mucosa [93]. Traditional vaginal drug delivery systems are unable to maintain an effective drug concentration for an extended period of time. In fact, antifungals for vaginal medications are required to be solubilized and retained at or near the mucosa for a long residence time and be able to penetrate the vaginal tissue to improve their therapeutic efficacy. Numerous investigations have reported that CDs act as solubilizers and permeation enhancers when incorporated into tablets, mucoadhesive gels, creams, and films for vaginal antifungal delivery. 

Natamycin is a poorly water-soluble compound. Solubility enhancement was achieved by complexation with γCD with a stability constant of 667 M^−1^ (Table 2). This complex provided a similar MIC when compared with the pure drug against *Candida*. Furthermore, the natamycin/γCD complex-loaded bioadhesive tablet, based on Carbopol 934P, provided good mucoadhesion as well as prolonged drug release [53]. Another case study of the itraconazole bioadhesive vaginal tablet was developed by Cevher et al. (2014) [94]. The aqueous solubility of itraconazole was greatly enhanced with SBEβCD, and the resultant complex exhibited the highest efficacy, i.e., the lowest MIC against *Candida albicans* among the tested complexes. Additionally, the itraconazole/SBEβCD complex incorporated in the Carbopol 934P-based bioadhesive tablet provided good mucoadhesion and prolonged release of itraconazole for 36 h.

Another polyene antibiotic, amphotericin B, is also therapeutically effective in vaginal candidiasis. However, its application is limited due to its poor aqueous solubility and toxicity. In the study of Kim et al. (2010) [95], they improved the aqueous solubility of amphotericin B using HPβCD or HPγCD. The amphotericin B/HPγCD complex, which demonstrated a significantly high solubility of amphotericin B, was further loaded in an in situ gel based on thermosensitive multiblock copolymers. These copolymers were synthesized from Pluronic^®^ triblock copolymers (Pluronic^®^ P85 and P104) and di-(ethylene glycol) divinyl ether. The developed formulation underwent a sol-gel transition at the site of action. Moreover, the developed amphotericin B/HPγCD complex-loaded in situ gel showed controlled-release characteristics and was less toxic than free amphotericin B in the HEK 239 cell line. In a histopathological study in mice, no visible signs of inflammation or necrosis in the vaginal mucosa were observed. 

Deshkar et al. (2019) [21] developed a voriconazole/HPβCD complex-loaded in situ gel for vaginal delivery. Due to the solubility of voriconazole being enhanced in the presence of HPβCD, a significantly higher drug release was observed compared with the voriconazole gel without HPβCD. A further in vivo vaginal tissue uptake study was conducted in Wistar rats. It was demonstrated that the HPβCD-based in situ gel formulation provided a higher drug uptake in the vaginal tissue than an in situ gel without the HPβCD and voriconazole dispersion. 

### 3.4. Pulmonary Drug Delivery

Pulmonary fungal infections are a major health problem, particularly in immunocompromised patients with lung cancer, cystic fibrosis, human immunodeficiency virus (HIV), etc. [96]. Conventional treatments in pulmonary delivery are limited due to several factors, including bypassing the upper airways to reach the lungs, the branching structure of the lungs resulting in difficult access to the alveolar regions, and the rapid blood turnover in the lungs. Another factor is the presence of a thick mucus gel layer in which the inhaled drugs could be entrapped and then removed by mucociliary clearance. As a result, a large dose is usually required for a long period of time to cause toxicity [96,97]. 

CDs were shown to improve the bioavailability of poorly water-soluble drugs in aerosols or solutions for pulmonary delivery [98]. Yang et al. (2010) [99] prepared an itraconazole/HPβCD complex solution for pulmonary delivery using nebulization. An itraconazole nanoparticle colloidal dispersion composed of mannitol and lecithin was developed using the ultra-rapid freezing method for comparison. Both the itraconazole/HPβCD solution and colloidal dispersion were suitable for deep lung drug delivery in mice. The itraconazole/HPβCD complex solution demonstrated a faster systemic absorption of itraconazole across the lung epithelium than that of a colloidal dispersion. This is possible because itraconazole is released from the nanoparticles and is required for the phase-to-phase transition. However, from the pharmacokinetic data, the nanoparticulate itraconazole colloidal dispersion exhibited an enhanced drug bioavailability because of the presence of lecithin as an additional permeation enhancer. The authors suggested that the uncertain stability and the regulatory status of HPβCD should be considered in pulmonary administration. 

Another research group approached treating invasive fungal infections by preparing a voriconazole/SBEβCD complex for targeted drug delivery to the lungs using nebulization [100]. The high solubilization capacity of voriconazole by SBEβCD in the developed aerosol solution resulted in high drug loading, leading to enhanced voriconazole concentrations in the lung tissues and plasma after single and multiple doses of inhaled administration. Hence, the pharmacokinetic data of inhaled aqueous solutions of voriconazole confirmed that effective therapeutic outcomes could be achieved using a voriconazole/SBEβCD solution.

### 3.5. Ocular Drug Delivery 

Ocular drug delivery is a considerable challenge in developing pharmaceutical formulations because of the complex anatomy of the eyes. The anatomical and physiological barriers in the eyes, such as tear turn over, nasolacrimal drainage and blinking of the eyes, etc., limit the bioavailability of drugs in ocular tissues. To obtain the optimum penetration, drugs need to be solubilized in lachrymal fluid and pass the tear film barrier [101]. The use of solubilizers, as well as permeation enhancers such as CDs could help to achieve therapeutic drug concentrations in various parts of ocular tissues. 

Mahmoud et al. (2011) [102] developed econazole/SBEβCD complex-loaded chitosan nanoparticles using the ionic gelation method. Ionically crosslinked chitosan/SBEβCD nanoparticles acted as an interesting chitosan-based delivery system for econazole to the ocular membrane. The developed formulations showed particle sizes in the nanometer range (90 to 673 nm) and positive zeta potential values (+22 to +33 mV). The optimum formulation provided good mucoadhesive properties in a controlled-release manner. The antifungal activity of the econazole-loaded chitosan/SBEβCD nanoparticles showed better efficacy when compared with an econazole solution. Regarding these observations, the authors concluded that the optimized formulation might exhibit a potential carrier for ocular drug delivery. 

A voriconazole/HPβCD complex-loaded hydrogel was developed by Diaz-Tome et al. (2021) to treat fungal keratitis [43]. The solubility of voriconazole was greatly increased using HPβCD. Furthermore, they prepared two types of gels, i.e., an ion-sensitive hydrogel prepared with a voriconazole/HPβCD-loaded kappa-carrageenan hydrogel and a Vfend^®^-loaded (a commercial intravenous voriconazole/SBEβCD preparation) hyaluronic acid hydrogel. The ex vivo permeability study showed that both hydrogels provided good corneal permeability. These formulations increased the residence time on the ocular surface and proved to be nonirritants. These formulations could serve as suitable alternatives to treat fungal keratitis.

Chaudhari et al. (2022) [40] formulated a ketoconazole/SBEβCD inclusion complex-loaded in situ gel for ocular fungal infections. The aqueous solubility of ketoconazole was enhanced by forming an inclusion complex with SBEβCD. The ketoconazole/SBEβCD complex was further loaded in a thermosensitive in situ gel. The drug release was increased 10-fold compared with the intact ketoconazole. The in situ gel exhibited a more sustained drug release because of the diffusion of drugs from the polymer matrix. The ex vivo permeation through a goat corneal membrane revealed that the permeation flux of ketoconazole from the formulation in the presence of SBEβCD (∼19.11 µg/cm^2^/h) was significantly higher than that in the absence of SBEβCD (∼1.17 µg/cm^2^/h). The formulations were nontoxic to human corneal endothelial cells and, thus, could potentially treat fungal keratitis. Another example is fluconazole/HPβCD complex-loaded Eudragit nanoparticles and chitosan-coated niosomal vesicles that are further incorporated into in situ gels [70]. The developed formulation resulted in nanosized particles with high positive zeta potentials and high entrapment efficiencies. It also exhibited sustained release within 24 h, enhancing the corneal permeation without irritating the ocular surface and promising antifungal activity against *Candida albicans*. The modified formulations using combined strategies, i.e., CD inclusion complexes, nanoparticulates, and in situ gels, provide the optimum eye drops for ocular delivery.

Pickering emulsions are surfactant-free emulsions stabilized by solid particles. Their unique structures endow them with good stability, excellent biocompatibility, and environmental friendliness. CD-based Pickering emulsions exhibited high safety and biocompatibility due to fewer quantities of stabilizers and were more stable against coalescence and separation when compared with conventional emulsions [103]. Recently, amphotericin B-loaded CD-based Pickering nanoemulsions were developed by Maw et al. (2022) [104]. This was achieved using αCD/medium chain triglyceride crystals as solid particle stabilizers at the oil and water interface. In addition, the solubility of amphotericin B was also increased by complexing with γCD and HPγCD. The developed formulations exhibited less amphotericin B aggregation and fewer hemolytic properties than commercial products (i.e., the amphotericin B-containing sodium deoxycholate micellar formulation). Moreover, sustained drug release was observed due to the high entrapment efficacy. The in vitro antifungal activity study also showed a better result in amphotericin B-loaded nanoemulsions than that of intact amphotericin B, and also comparable with that of commercial products against *Candida albicans*. The developed CD-based Pickering nanoemulsions were physically and chemically stable over six months, which was greater than their respective non-Pickering nanoemulsions. 

### 3.6. Dermal Drug Delivery

Topical antifungal drug delivery is one of the major routes to treat skin diseases associated with fungi. The route ensures direct access and a higher retention rate at the infected area on the skin [105]. Furthermore, topical delivery also decreases systemic side effects and avoids first-pass metabolism. Che et al. (2015) [106] developed a CD-based microemulsion loaded with ketoconazole for skin delivery. In the comparative study with the microemulsion without CD, the ketoconazole/HPβCD complex, ketoconazole aqueous suspensions, as well as the marketed product (ketoconazole cream, Daktarin Gold^®^), the incorporation of HPβCD in the microemulsions provided a better retention time and enhanced permeation through the skin layer. The synergistic skin targeting effect might have been due to the high solubilization effect of HPβCD, which was able to promote drug deposition within the skin, as well as drug penetration through the skin. The in vitro antifungal activity study revealed that ketoconazole/HPβCD complex-loaded microemulsions resulted in the largest zone of inhibition against *Candida parapsilosis*. Therefore, it could be concluded that a microemulsion combination with HPβCD may be a promising approach for the skin-targeted delivery of ketoconazole. 

Deformable liposomes have also emerged to improve their effectiveness as drug carriers to increase drug penetration through biological mucous membranes due to their high flexibility and elasticity. These systems can squeeze themselves between the stratum corneum cells and reach deep skin layers. Itraconazole/HPβCD-loaded deformable liposomes were investigated by Alomrani et al. (2014) [107]. HPβCD was incorporated to increase the solubility of drugs and to enhance skin penetration. The presence of CD provided physical stability to the developed liposome. In addition, higher deformability was observed, leading to good permeation towards the deep skin layer when compared with conventional liposomes. The in vitro antifungal activity revealed that the deformable liposome was active against *Candida albicans*. 

The potential of CD-based Pickering emulsions as a carrier of econazole for topical drug delivery was introduced by Leclercq and Nardello-Rataj (2016) [108]. An econazole-loaded Pickering emulsion was developed using αCD, βCD, and γCD with paraffin oil and isopropyl myristate. Of these, αCD resulted in the most stable Pickering emulsion, followed by βCD and γCD due to the suitable affinity between the hydrophobic alkyl tail of the oil molecule and the small αCD cavity. The resulting Pickering emulsions were stable against the coalescent due to the formation of a dense film of oil/CD complex particles at the oil and water interface. The antifungal activity results displayed that the Pickering emulsions stabilized with αCD and βCD were able to inhibit *Candida albicans* growth. 

Recently, CD nanosponge, a novel nanotechnology platform containing CD was applied in the pharmaceutical field.. CD nanosponges are nanostructured materials with nanometer-sized cavities that are easily prepared using crosslinkers, such as carbonyl diimidazole or diphenyl carbonate [109,110]. Srivastava et al. (2021) [111] developed an econazole-loaded nanosponge comprising βCD and N,N-carbonyl diimidazole as a crosslinker. The optimized econazole nanosponge was further loaded in a topical hydrogel in which Carbopol 934P was used as a gelling agent. The developed nanogel containing the econazole-loaded nanosponge achieved a higher penetration of econazole across the skin barrier with a flux of 314 µg/cm^2^ than the commercial product with a flux of 88 µg/cm^2^ during the 24 h study. The nanogel showed better in vitro antifungal activity against *Candida albicans* and in vivo antifungal activity in male albino Wistar rats than the commercially marketed product. These promising results might have been due to the function of the nanosponge, which helped econazole to permeate the skin layer and provided the controlled release of econazole for longer periods of time during the study.

### 3.7. Parenteral Drug Delivery

Invasive fungal infections have been recognized as a major cause of morbidity and mortality in immunocompromised patients. The parenteral route is preferred for the delivery of compounds with a narrow therapeutic index and poor bioavailability and for the prescription of unconscious patients. To maintain therapeutic effectiveness, frequent injections are required, leading to patient discomfort. Therefore, for parenteral antifungal drug delivery, developing formulation technologies is considered important to provide a targeted therapeutic level and sustained release of drugs in a predictable manner [112]. 

Posaconazole is a triazole antifungal agent with an extended spectrum of antifungal activity. However, its pharmacological effects are limited owing to its poor aqueous solubility and low bioavailability. The enhancement of the bioavailability of posaconazole was obtained with the complexation with SBEβCD, reported by Wang et al. (2018) [113]. The solid posaconazole/SBEβCD complex, prepared by lyophilization, showed an excellent dissolution profile within 5 min. In addition, in vivo pharmacokinetic parameters were studied in rats after pure posaconazole was solubilized in cosolvents, and posaconazole/SBEβCD complexes were injected intramuscularly. The data showed that the posaconazole/SBEβCD complex significantly increased the peak concentration and bioavailability compared with pure posaconazole. The authors suggested that SBEβCD played an important role in stabilizing the saturated drug solution and preventing the precipitation of drugs at the injection site.

Mutlu-Agardan et al. (2020) [114] studied an amphotericin B and amphotericin B/αCD complex double-loaded liposome to treat invasive fungal infections. Among the CDs tested, the highest solubilization was achieved with αCD, and this complex was further loaded into liposome preparations. The drug-in-CD-in-liposomes improved the physical stability and exhibited a higher percentage of entrapment efficiency, i.e., 80%, than only free amphotericin B-incorporated liposomes. Additionally, the novel amphotericin B double-loaded liposome formulation provided a rapid onset time followed by a sustained release property within 72 h. The antifungal activity against *Candida albicans* showed that the amphotericin B double-loaded liposome exhibited significantly lower MIC and minimum fungicidal concentration (MFC) values when compared with the conventional and commercial products. The antifungal drug-loaded CD-based formulations in different routes of administration and their findings are summarized in Table 4.

In general, CDs are used to enhance the aqueous solubility of antifungals and to overcome the limitations of nanoparticulate drug delivery systems, which include improving drug loading and entrapment efficiency. Another beneficial finding regarding CD-containing oral solid dosage forms is the enhancement of drug dissolution in aqueous media. Moreover, CD aids in decreasing the degradation of labile drugs and preserving antifungal potencies. Regarding toxicological issues, the drug-induced hemolysis of, for example, amphotericin B was significantly lowered with CD complexation. Additionally, it was noted that the incorporation of CD in formulations resulted in modified drug release, enhanced drug permeation through biological membranes, improved pharmacokinetics, and consequently increased the bioavailability of antifungals.

## 4. Conclusions

In this review, we discussed how the unique properties of CDs could be applied to increase the solubility of antifungal drugs, as well as how water-soluble antifungal/CD complexes are incorporated into various nanocarriers. Antifungal/CD complex-loaded nanoparticulate platforms have emerged as effective nanosystems for antifungal agents for different routes of administration. They have significantly improved the drug loading, permeation, and chemical stability of drugs compared to systems without CDs. The increased bioavailability of antifungal agents has resulted in reduced dosing frequencies. At present, researchers have attempted to modify CDs and nanocarriers and combine several approaches to create more complex nanosystems. Optimized antifungal formulations have been successfully developed that have improved drug bioavailability and antifungal efficacies. Apart from the promising results of CD-based nanocarriers, the long-term safeties, toxicological profiles, and regulatory statuses associated with the excretion of nanomaterials require further evaluation.

## Figures and Tables

**Figure 1 pharmaceuticals-15-01447-f001:**
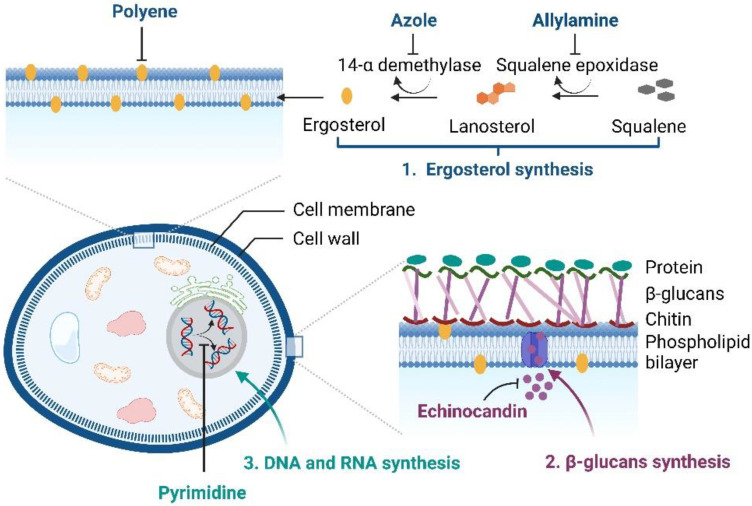
Mechanisms of the actions of antifungal drugs (created with Biorender.com).

**Figure 2 pharmaceuticals-15-01447-f002:**
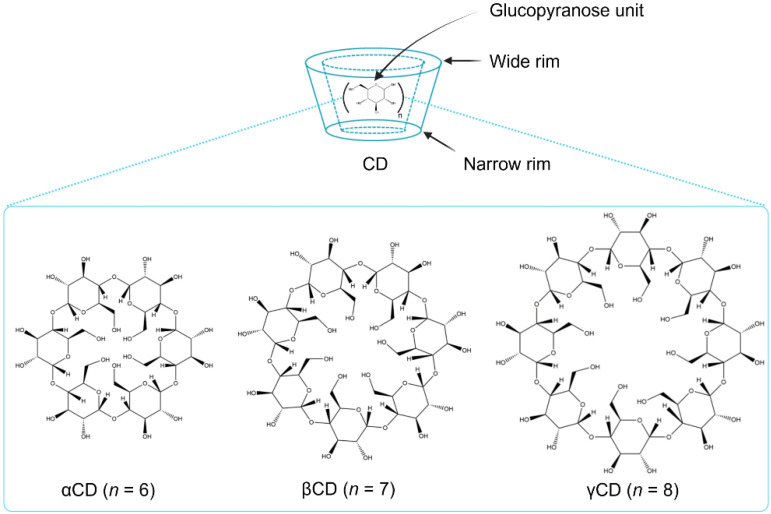
Schematic presentation of the parent CDs (created with Biorender.com).

**Figure 3 pharmaceuticals-15-01447-f003:**
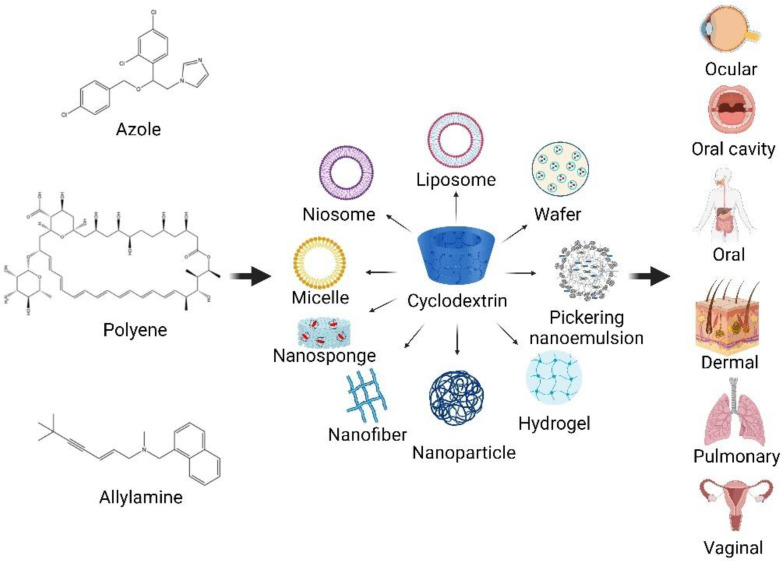
CD-based nanocarriers for antifungal agents in drug delivery systems (created with Biorender.com).

**Table 1 pharmaceuticals-15-01447-t001:** Physicochemical properties of natural CDs and selected CD-derivatives of pharmaceutical interest.

Cyclodextrin	Number of Substituents ^a^	MW (g/mol) of Given Substitution (MS) ^b^	Solubility in Water (mg/mL) ^c^	Calculated LogP_(octanol/water)_ at 25 °C	Refs.
α-cyclodextrin (αCD)	-	972	145	−13	[24,25,26]
β-cyclodextrin (βCD)	-	1135	18.5	−14	[24,25]
γ-cyclodextrin (γCD)	-	1297	232	−17	[24,25]
2-hydroxypropyl-αCD (HPαCD)	3.6	1199 (0.65)	>500	<−10	[24,27]
Randomly methylated-βCD (RMβCD)	9.7–13.6	1312 (1.8)	>500	−6	[24,25,28]
2-hydroxypropyl-βCD (HPβCD)	2.8–10.5	1400 (0.65)	>600	−11	[24,25,29]
Hydroxyethyl-βCD (HEβCD)	3.6	1443 (1.0)	>2000	-	[24,29,30]
Sulfobutylether-βCD (SBEβCD)	6.2–6.9	2163 (0.9)	>500	<−10	[24,25]
2-hydroxylpropyl-γCD (HPγCD)	3.0–5.4	1576 (0.6)	>500	−13	[24,25]

^a^ The average number of substituents per one CD molecule. ^b^ MW = the molecular weight of the unhydrated CD; MS = the molar degree of substitution is defined as the average number of substituents that have reacted with one glucopyranose repeat unit. ^c^ the solubility in pure water at approximately 25 °C.

**Table 2 pharmaceuticals-15-01447-t002:** Summarized data of the stability constant (K_1:1_ and K_1:2_) and complexation efficiency (CE) values of antifungal/CD complexes.

Drug	CD ^a^	Medium	pH	Type	K_1:1_ (M^−1^)	K_1:2_ (M^−1^)	CE	Refs.
Miconazole	αCD	Water (25 °C)	-	A_L_	333	-	-	[32]
		Water (25 °C)	-	A_L_	436	-	0.21	[33]
	βCD	Water (25 °C)	-	A_L_	293	-	-	[32]
		Water (25 °C)	-	B_S_	596 ^b^	-	0.29	[33]
		Water (37 °C)	-	A_L_	6065	-	0.902	[34]
		Phosphate buffer	pH 6	A_L_	97	-	-	[35]
			pH 7	A_L_	82	-	-	
			pH 8	A_L_	65	-	-	
			pH 9	A_L_	39	-	-	
	γCD	Water (25 °C)	-	A_L_	695	-	-	[32]
		Water (25 °C)	-	A_N_	488	-	0.24	[33]
	HPβCD	Water (25 °C)	-	A_L_	363	-	-	[32]
		Water (37 °C)	-	A_L_	1017	-	0.361	[34]
	HPγCD	Water (25 °C)	-	A_L_	305	-	-	[32]
	HEβCD	Water (25 °C)	-	A_L_	312	-	-	[32]
Econazole	αCD	Water (22–23 °C)	pH 3	A_L_	354.5	-	0.371	[36]
			pH 5	A_L_	2597.5	-	0.293	
			pH 7.5	A_P_	870.2	15.0	0.041	
		Water (25 °C)	-	A_L_	505.54	-	0.075	[37]
		Phosphate buffer saline (25 °C)	pH 7.4	A_L_	6.28	-	0.036	
	γCD	Water (22–23 °C)	pH 3	B_S_	246.7 ^b^	-	0.258	[36]
			pH 5	B_S_	1032.8 ^b^	-	0.117	
			pH 7.5	- ^c^	- ^c^	- ^c^	- ^c^	
	HPβCD	Water (25 °C)	-	A_L_	52.21	-	0.077	[37]
		Phosphate buffer saline (25 °C)	pH 7.4	A_L_	86.34	-	0.051	
	HPγCD	Water (25 °C)	-	A_L_	54.11	-	0.080	[37]
		Phosphate buffer saline (25 °C)	pH 7.4	A_L_	40.35	-	0.025	
Ketoconazole	βCD	Water (25 °C)	-	A_L_	1859	-	-	[38]
		Water (37 °C)	-	A_L_	4966	-	-	[39]
	SBEβCD	Water	-	A_L_	843	-	0.399	[40]
Itraconazole	βCD	Phosphate buffer (25 °C)	pH 7.4	A_L_	885	-	-	[41]
	HPβCD	Water (25 °C)	pH 2	A_P_	5280	38	-	[42]
			pH 4	A_P_	15	2504	-	
			pH 7	A_P_	1926	1	-	
Voriconazole	αCD	Water (25 °C)	-	A_L_	55.14	-	0.07	[43]
	HPβCD	Water (25 °C)	-	A_L_	224.27	-	0.29	[43]
		Water (37 °C)	-	A_L_	320	-	-	[44]
	SBEβCD	Water (25 °C)	-	A_L_	324.98	-	0.741	[45]
	HPγCD	Water (25 °C)	-	A_L_	242.65	-	0.32	[43]
Posaconazole	βCD	Water (15 °C)	-	A_P_	296.66	935.42	-	[46]
		Water (25 °C)	-	A_P_	300.28	983.19	-	
		Water (37 °C)	-	A_P_	307.12	1025.44	-	
	HPβCD	Water (15 °C)	-	A_P_	441.23	1289.18	-	[47]
		Water (25 °C)	-	A_P_	494.67	1337.24	-	
		Water (37 °C)	-	A_P_	431.36	1385.47	-	
	DMβCD	Water (15 °C)	-	A_P_	393.25	1269.53	-	[46]
		Water (25 °C)	-	A_P_	398.13	1296.27	-	
		Water (37 °C)	-	A_P_	405.86	1340.29	-	
Amphotericin B	αCD	Water (25 °C)	-	A_L_	146	-	0.002	[48]
	βCD	Water (25 °C)	-	A_L_	72.1	-	0.001	[48]
	γCD	Water (25 °C)	-	A_P_	4972.3	14.1	0.069	[48]
		Water (25 °C)	-	A_P_	462	42	-	[49]
		Water (25 °C)	-	A_L_	1129	-	0.016	[50]
	HPγCD	Water (25 °C)	-	A_P_	2851.7	17.0	0.039	[48]
Nystatin	αCD	Water (25 °C)	-	B_S_	- ^c^	- ^c^	- ^c^	[51]
	βCD	Water (25 °C)	-	A_L_	0.375	-	-	[51]
	γCD	Water (25 °C)	-	A_L_	0.539	-	-	[51]
Natamycin	βCD	Water (25 °C)	-	A_L_	1010	-	-	[52]
	γCD	Water (25 °C)	-	A_N_	- ^c^	- ^c^	- ^c^	[52]
		Water (25 °C)	-	A_L_	667	-	-	[53]
	HPβCD	Water (25 °C)	-	A_N_	- ^c^	- ^c^	- ^c^	[52]
Flucytosine	βCD	Water (22 °C)	-	A_L_	70	-	-	[54]
	HPβCD	Water (22 °C)	-	A_L_	297	-	-	[54]
Terbinafine	αCD	0.05 M Disodium hydrogen phosphate and 1M of Sodium hydroxide (25 °C)	pH 12	A_P_	2.8	1.1	-	[55]
	βCD	0.05 M Disodium hydrogen phosphate and 1M of Sodium hydroxide (25 °C)	pH 12	B_S_	25	-	-	[55]
	γCD	0.05 M Disodium hydrogen phosphate and 1M of Dodium hydroxide (25 °C)	pH 12	A_L_	0.66	-	-	[55]
	HPβCD	0.05 M Disodium hydrogen phosphate and 1M of Dodium hydroxide (25 °C)	pH 12	A_L_	23	-	-	[55]
	MβCD	0.05 M Disodium hydrogen phosphate and 1M of Dodium hydroxide (25 °C)	pH 12	A_P_	46	-	-	[55]

^a^ αCD: α-cyclodextrin; βCD: β-cyclodextrin; γCD: γ-cyclodextrin; HPβCD: hydroxypropyl β-cyclodextrin; MβCD: methyl β-cyclodextrin; SBEβCD: sulfobutyl ether β-cyclodextrin; HEβCD: hydroxyethyl β-cyclodextrin; DMβCD: dimethyl β-cyclodextrin; HPγCD: hydroxypropyl γ-cyclodextrin; ^b^ calculated from the linear part of the phase-solubility diagram, ^c^ could not be determined.

**Table 3 pharmaceuticals-15-01447-t003:** Observations of CDs enhancing the permeability of antifungals through biological membranes in vitro, ex vivo, and in vivo studies.

Cyclodextrin	Drug	Biological Membrane	Observations	Refs.
βCD	Itraconazole	Rabbits’ cornea	Significantly higher ex vivo corneal flux compared with the drug suspension.	[69]
HPβCD	Fluconazole	Rabbits’ cornea	Enhanced in vivo permeation compared with the plain drug solution.	[70]
SBEβCD	Ketoconazole	Goats’ cornea	Increased ex vivo corneal permeation compared with ketoconazole alone	[40]
RMβCD	Itraconazole	Caco-2 cell (intestinal cell line) monolayer	Enhanced in vitro permeation compared with drugs solubilized in dimethyl sulfoxide	[20]
HPβCD	Voriconazole	Wister rats’ vaginal mucosa	Provided higher in vivo vaginal tissue uptake than without the HPβCD and voriconazole dispersion.	[21]

**Table 4 pharmaceuticals-15-01447-t004:** Examples of CD-based formulations containing antifungal agents for various drug delivery systems.

Route of Administration	Drug	CD	Dosage Form/Delivery System	Experimental Findings	Refs.
Oral	Itraconazole	RMβCD	Floating tablet	Increased the drug solubility in the complexing medium, pH 1.2.; zero-order kinetic releasing profile, increased the floating time in the stomach, and increased the oral bioavailability.	[20]
	Posaconazole	HPβCD	Inclusion complex	Significantly increased the aqueous solubility of drugs, enhanced the drug dissolution rate in different simulated gastrointestinal conditions, and showed good susceptibility to microorganisms, i.e., *Candida*, *Aspergillus*, and *Penicillium*.	[47]
Oral local cavity	Econazole	SBEβCD	Wafer	Increased the drug solubility and dissolution rate; demonstrated controlled-release properties and mucoadhesive and antifungal efficacy.	[85]
	Miconazole	HPβCD	Chewing gum	Demonstrated high drug release and enhanced antifungal activity	[87]
	Clotrimazole	HPβCD	Nanofiber	Demonstrated high mucoadhesive properties, an initial fast release followed by controlled release, a fast therapeutic efficacy in antifungals, and biocompatibility	[91]
Ocular	Econazole	SBEβCD	Nanoparticle	Good mucoadhesive and controlled-release characteristics and antifungal efficacy	[102]
	Fluconazole	HPβCD	Nanoparticle loaded in an situ gel and a noisome loaded in situ gel	Sustained release, enhanced corneal permeation, nonirritant to the ocular surface, and promising antifungal activity against *Candida albicans*	[70]
	Itraconazole	βCD	Micelle	Good stability, safe, mucoadhesive and ability to permeate through the cornea of a rabbit, and reduced clinical symptoms in the anterior chamber, white lesions, and corneal opacity	[69]
	Amphotericin B	γCD, HPγCD	Pickering nanoemulsion	Less amphotericin B aggregation and hemolytic properties than commercial products, sustained in vitro drug release, better stability than conventional nanoemulsion, and active against *Candida albicans*	[104]
	Voriconazole	HPβCD	In situ gel	High mucoadhesive property, enhanced ex vivo permeation, and biocompatibility	[43]
	Ketoconazole	SBEβCD	In situ gel	Increased ex vivo corneal permeation, increased corneal retention time, and biocompatibility	[40]
Pulmonary	Voriconazole	SBEβCD	Aerosol solution	Increased solubility, high bioavailability in lung tissue, and plasma	[100]
Vaginal	Itraconazole	SBEβCD	Tablet	Increased solubility and increased antifungal efficacy, bioadhesive property, and prolonged drug release	[94]
	Amphotericin B	HPγCD	In situ gel	Increased solubility, biocompatibility, and controlled-release property	[95]
	Voriconazole	HPβCD	In situ gel	Mucoadhesive property, sustained release, and increased in vivo vaginal tissue uptake	[21]
	Natamycin	γCD	Bioadhesive tablet	High mucoadhesion and prolonged drug release	[53]
Dermal	Itraconazole	HPβCD	Deformable liposome	Better permeability to the skin layer than conventional liposomes and active antifungal activity against *Candida albicans*	[107]
	Econazole	βCD	Nanosponge	Enhanced permeation in in vitro goatskin, ability to inhibit fungal growth in both in vitro and in vivo studies	[111]
	Econazole	αCD, βCD, γCD	Pickering emulsion	Good stability, biocompatibility, and antifungal activity against *Candida albicans*	[108]
Systemic	Posaconazole	SBEβCD	Inclusion complex	Enhanced solubility and dissolution property, improved bioavailability (1.6 times higher) than pure posaconazole	[113]
	Amphotericin B	αCD	Double-loaded liposome	Higher physical stability, controlled-release manner, and four times higher antifungal activity (minimal MIC and MFC values) than the marketed product Ambisone^®^	[114]

## Data Availability

Data sharing not applicable.

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
