# Peer review of "A Current Overview of Cyclodextrin-Based Nanocarriers for Enhanced Antifungal Delivery"

_pharmaceuticals, 2022, doi:10.3390/ph15121447_

Round 1

Reviewer 1 Report

In the manuscript Current Overview of Cyclodextrin-Based Nanocarriers for Enhanced Antifungal Delivery, the author summarized the application of CD in antifungal drug formulations. This is a nice organized review article. The review works done by the authors were informative and valuable. I would recommend accepting the paper after modifications. I have some comments to the authors.

On the whole, the author only introduced the corresponding research, and the author's own opinions are lacking in the manuscript. Since it is a review article, more discussion should be added. It is recommended that the author add critical reviews and comparisons on related studies. 

Section 1 seems to be odd and doesnt provide valuable information. It is recommended not to talk too much about fungal infections in humans mechanisms of action of antifungals. It might be better to remove table 1. It might be better to appropriately mention the use of cyclodextrin inclusion complexes in other drugs or natural active substances.

CD-based nanocarriers in other enhanced antifungal delivery,such as fungicides for agricultural use and natural fungicide, should be mentioned in the first paragraph of Section 3. https://doi.org/10.1016/j.indcrop.2021.114300

https://doi.org/10.1021/acs.jafc.2c01866

https://doi.org/10.1021/acs.jafc.1c01351

References were mainly early studies. Recent studies should be added.

Reviewer 2 Report

This review describes the application cyclodextrins and their combination with nanoparticulate systems to antifungal drug formulations for various routes of administration. The review is well and comprehensively documented and will provide useful information in the antifungal delivery field. My minor comments are as follows:

1. The solubility of the parent CDs (lines 90-93): The aqueous solubility of bCD is very low as described, but those of aCD and gCD are not so low. Please comment on the solubility difference.

2. Table 2: Please check the degree (3.8) of substitution of dimethyl-bCD (DMbCD). The maximum number of the degree of substitution is 3 due to the presence of three hydroxyl groups in one glucose unit.

3. Table 2: The degree of substitution of hydroxyethyl-bCD (HEbCD) was described to be 1.0 in the table. Is that a single component? It is better to show the reference for each cyclodextrin.
